# Explaining *Echis*: Proteotranscriptomic Profiling of *Echis carinatus carinatus* Venom

**DOI:** 10.3390/toxins17070353

**Published:** 2025-07-16

**Authors:** Salil Javed, Prasad Gopalkrishna Gond, Arpan Samanta, Ajinkya Unawane, Muralidhar Nayak Mudavath, Anurag Jaglan, Kartik Sunagar

**Affiliations:** Evolutionary Venomics Lab, Centre for Ecological Sciences, Indian Institute of Science, Bangalore 560012, Karnataka, India; javedvpjaved@gmail.com (S.J.); prasadgond1@iisc.ac.in (P.G.G.); arpansamanta@iisc.ac.in (A.S.); ajinkyau@iisc.ac.in (A.U.); muralidharn@iisc.ac.in (M.N.M.); anuragjaglan25@gmail.com (A.J.)

**Keywords:** *Echis carinatus carinatus*, transcriptome, proteome, antivenom, ED_50_

## Abstract

Snakebite remains the most neglected tropical disease globally, with India experiencing the highest rates of mortality and morbidity. While most envenomation cases in India are attributed to the ‘big four’ snakes, research has predominantly focused on Russell’s viper (*Daboia russelii*)*,* spectacled cobra (*Naja naja*)*,* and common krait (*Bungarus caeruleus*), leading to a considerable gap in our understanding of saw-scaled viper (*Echis carinatus carinatus*) venoms. For instance, the venom gland transcriptome and inter- and intra-population venom variation in *E. c. carinatus* have largely remained uninvestigated. A single study to date has assessed the effectiveness of commercial antivenoms against this species under in vivo conditions. To address these crucial knowledge gaps, we conducted a detailed investigation of *E. c. carinatus* venom and reported the first venom gland transcriptome. A proteotranscriptomic evaluation revealed snake venom metalloproteinases, C-type lectins, L-amino acid oxidases, phospholipase A_2_s, and snake venom serine proteases as the major toxins. Moreover, we assessed the intra-population venom variation in this species using an array of biochemical analyses. Finally, we determined the venom toxicity and the neutralising efficacy of a commercial antivenom using a murine model of snake envenoming. Our results provide a thorough molecular and functional profile of *E. c. carinatus* venom.

## 1. Introduction

As a genus whose distribution spans across continents, *Echis* is responsible for a large proportion of clinically relevant snakebites in Africa, the Middle East, India, Pakistan, and Sri Lanka [1]. Comprising at least twelve species described to date, the members of this genus are known as saw-scaled or carpet vipers, given the presence of body scales that are keeled at an angle of 45° [2]. In India, this genus is represented by two sister sub-species: the saw-scaled viper (*Echis carinatus carinatus*) and Sochurek’s viper (*Echis carinatus sochureki*) [3]. The saw-scaled viper (*E. c. carinatus*) is a member of the so-called ‘big four’ Indian snakes that include Russell’s viper (*Daboia russelii*), spectacled cobra (*Naja naja*), and common krait (*Bungarus caeruleus*). Together, these snakes cause the highest number of snakebites in the country, accounting for 58,000 annual deaths [4]. Envenoming by *E. c. carinatus* is entailed by anomalies in the haemostatic system, such as oedema, blister formation, myonecrosis, hypertension, and other systemic coagulopathies [5]. Infestation of acute kidney injury (AKI) has also been reported in bite victims, wherein pleiotropic effects, such as the constriction of glomeruli, the collapse of the capillary lumen, and degradation of the columnar epithelial layer in nephrons, lead to complete renal failure [6]. However, despite their enormous clinical relevance, both sister sub-species of saw-scaled vipers remain some of the most poorly investigated snakes from the Indian subcontinent. For instance, only a handful of studies have determined the venom compositions of saw-scaled vipers [7,8,9,10,11] and their venom gland transcriptome remains undescribed. Only two studies have evaluated the effectiveness of antivenoms in neutralising the venoms of *E. c. carinatus* in the mouse model of snake envenoming [10,11].

To address the aforementioned shortcomings and to advance our understanding of saw-scaled viper venoms, we implemented a multi-faceted approach. We sequenced the venom gland transcriptome of *E. c. carinatus* for the first time, whilst also characterising its venom proteome. The proteotranscriptomic approach implemented in this study enabled the identification of the major components of *E. c. carinatus* venom. This was followed by the evaluation of the biochemical activities of these major toxins, which highlighted a considerable intra-population venom variation. Finally, we assessed the in vitro binding efficacy and in vivo neutralising potency of commercial Indian antivenoms against *E. c. carinatus* venom.

## 2. Results and Discussion

### 2.1. Comparative Venom Gland Transcriptomics

Despite being one of the ‘big four’ snake species in India, the venom profile of *E. c. carinatus* has been relatively poorly studied. While some studies have reported the venom proteome of *E. c. carinatus* [7,8,10], the venom gland transcriptome remains uninvestigated. To address this knowledge gap, we sequenced both venom glands (VGL and VGR) and a physiological tissue (muscle or intestine) of *E. c. carinatus* on an Illumina Novaseq 6000 platform, which generated between 30,135,873 and 80,757,205 reads (Appendix A). The de novo super assembly for EcCaKA05 and EcCaKA22 samples, constructed using venom gland and physiological tissue transcriptomes, resulted in the reconstruction of 587,908 (N50: 1424) and 735,540 transcripts (N50: 580), respectively (Appendix A). The completeness of the assembly as assessed by BUSCO (Sauropsida dataset) revealed that 84.92% and 83.13% of the groups had complete gene representation (single-copy or duplicated), while 2.74% and 4.47% were partially recovered, and 12.41% and 12.34% were missing. Transcript-level quantification for VGL, VGR, and the physiological tissue revealed high overall alignment rates (Appendix A).

The venom gland transcriptome of *E. c. carinatus*, being described here for the first time, was found to be similar to the previously described venom gland transcriptomes of other *Echis* species [12,13]. The annotation and differential expression analysis identified several toxin transcripts in both *Echis* venom gland transcriptomes. Nearly half of these transcripts were snake venom metalloproteinase (SVMP), ranging from 51.4% to 61.9% (Figure 1A). The other major components were C-type lectins or Snaclecs (CTL: 11.9–26%) and L-amino acid oxidase (LAAO: 6.2–11.8%). Phospholipase A_2_ (PLA_2_) showed a large variation between the two samples, ranging from 0.2% to 9.2%. Other proteases, such as snake venom serine protease (SVSP: 3.4–5.7%) and snake venom aspartic protease (SVAP: 0.6–2.2%), were also documented. Interestingly, inhibitors of many of these toxins, such as SVMPI, SVSPI, and PLA_2_I, were also detected in the transcriptome. The transcripts for disintegrin, vascular endothelial growth factor (VEGF), phospholipase B (PLB), 5′-nucleotidase (5′-NT), three-finger toxin (3FTx), cobra venom factor (CVF), phosphodiesterase (PDE), neprilysin, and hyaluronidase were also recovered (Appendix A), albeit in lower amounts (<1%). Overall, the expression profiles of both venom glands were very similar.

### 2.2. Venom Proteome

RP-HPLC and in-gel or in-solution digestion (Appendix A) followed by mass spectrometry analysis yielded 156 venom components belonging to 21 distinct families (File S2). The venom proteome of E. c. carinatus (EcCaKA08) was largely consistent with the literature [7,8,10], as well as the venom gland transcriptome herein presented, in terms of major toxins present. It was dominated by SVMP (39%), CTL (20.1%), SVSP (20%), and PLA_2_ (11.9%) (Figure 1B). The only exception was the detection of SVSPs in relatively larger amounts compared to the previous description of E. c. carinatus venom proteome [7,8,10]. An increase in the translation of SVSPs and PLA_2_s compared to their transcription (4 times and 1.5 to 5 times, respectively) was noted. This could point to a differential regulation for these toxins at the level of translation. We also detected LAAO (3.5%) and disintegrin (2.7%) in the venom proteome, but in lower amounts. SVMPI (0.84%)—identified in the E. c. carinatus venom proteome for the first time—and SVAP (0.28%) were found in trace amounts (Figure 1B). Acetylcholinesterase (AChE), nerve growth factor (NGF), and cysteine-rich secretory proteins (CRISPs) were exclusively identified in the proteome. On the contrary, we did not find neprilysin, cathepsin, waprin, or carboxypeptidase D in the proteome despite recovering transcripts for them.

### 2.3. Biochemical Activities of Venom

While the biochemical characterisation of *E. c. carinatus* venom has been reported, it has typically involved pooled venoms or individual samples from point locations [7,8,10]. Therefore, we evaluated the intra-population variations in the biochemical profiles of *E. c. carinatus* by sourcing multiple samples from Southern India.

#### 2.3.1. Phospholipase Activity of Venoms

PLA_2_s, one of the major toxin families in snake venom [14], are known to cause severe cytotoxic, hemotoxic, proinflammatory, and edematogenic symptoms in snakebite victims [15,16]. Considering their medical relevance, we investigated the phospholipase activity of *E. c. carinatus* venoms by incubating them with a chromogenic lipid substrate (i.e., NOB), followed by the measurement of absorbance at 425 nm for 40 min. A significant intra-population difference was observed in the phospholipase activity (*p* < 0.001; F-value (degrees of freedom between group (DFn), degrees of freedom within group (DFd)) = 59.96 (3, 8); Figure 2A). The highest activity was displayed by EcCaKa08 (98.09 nmol/mg/min). This was closely followed by EcCaKa10 (94.43 nmol/mg/min) and EcCaKa07 (82.30 nmol/mg/min). The lowest activity was recorded for EcCaKA09 (40.37 nmol/mg/min).

#### 2.3.2. LAAO Activity of Venoms

LAAO, an important enzymatic component in viperid snake venoms, induces cytotoxicity, oedema, and haemorrhage, and can also induce or inhibit platelet aggregation [17,18,19]. To assess the LAAO activity of *E. c. carinatus* venoms, 0.5 μg of the respective sample was incubated with the L-leucine substrate for 10 min. All individuals evaluated in this study exhibited a high LAAO activity, and significant differences (*p* < 0.001; F (DFn, DFd) = 40.91 (3, 8)) were observed across samples (between 5185.45 and 7609.70 nmol/mg/min; Figure 2B).

#### 2.3.3. Protease Activity of Venoms

Snake venom proteases, particularly SVSPs and SVMPs, are abundantly secreted in the venoms of viperid snakes [14]. They are well-known to inflict fibrinogenolytic, haemorrhagic, and cytotoxic activities [15,20]. Therefore, to understand the proteolytic activity of *E. c. carinatus* venoms, a protease assay was conducted using azocasein as a substrate. Compared with the bovine pancreatic protease positive control, all venoms exhibited a relatively moderate activity, ranging between 33.6% to 40% (Figure 2C). The proteolytic activity also showed a significant intra-population variation (*p* < 0.001; F (DFn, DFd) = 4297 (3, 32)). The lowest activity was detected in the individual EcCaKA07 (33.6%), followed by EcCaKA09 (34.1%) and EcCaKA08 (37.8%), while the highest activity was recorded for EcCaKA10 (40%).

To determine whether the proteolytic activity of *E. c. carinatus* venom is driven by SVSP or SVMP, the venoms were treated with PMSF and EDTA to inhibit SVSP and SVMP, respectively. This resulted in a significant reduction in activity in all venoms (*p* < 0.001). While the inhibition of both SVSP and SVMP was shown to reduce the proteolytic activity, the effect of inhibiting SVMP was much higher (*p* < 0.0001). These results suggest that SVMP is largely responsible for the proteolytic activity of *E. c. carinatus* venom. Interestingly, although EDTA reduces most of the proteolytic activity, the addition of EDTA and PMSF together completely abolishes this activity in three of the four venoms tested (EcCaKA08, EcCaKA09, and EcCaKA10, *p* < 0.0001; Figure 2C). This further points to the involvement of SVSP in proteolysis, albeit in a limited capacity.

The venom’s biochemical activity profile is consistent with its proteome composition (EcCaKA08). The major contribution of snake venom metalloproteinases (SVMPs) to the proteolytic activity is therefore expected, as they are the most abundant component. Overall, the results of our biochemical assessments suggest that intra-population variation in phospholipase, LAAO, and proteolytic activities could influence the clinical outcomes of *E. c. carinatus* bite victims.

### 2.4. The Preclinical Efficacy of Indian Antivenom Against E. c. carinatus Venom

Despite being one of the ‘big four’ snakes of India, *E. c. carinatus* venom is the least studied. Only a few studies have evaluated venom variation in *E. c. carinatus* and the differential immunorecognition potential of commercial Indian antivenoms against this venom. Furthermore, while the antivenom efficacy has been evaluated through in vitro assays [21], only two studies have determined the neutralising potency of antivenoms against *E. c. carinatus* venom in a murine model of envenoming [10,11].

#### 2.4.1. In Vitro Binding Potential

In vitro binding assays were performed to determine the immunorecognition potential of Indian polyvalent antivenoms (Premium Serums, VINS Bioproduct, Bharat Serums, Biological E. and Haffkine BioPharmaceutical) against the pooled venom of *E. c. carinatus*. The polyvalent antivenoms are raised against the ‘big four’ snakes of India. Endpoint titration ELISA was performed by incubating different dilutions of antivenom with a specific amount of venom to ascertain the binding efficacy. The absorbance measured at 405 nm was plotted against the respective dilution. All antivenoms exhibited higher binding efficacy than the negative control (*p* < 0.0001; Figure 3). However, at the highest dilution, Premium Serums antivenom exhibited the highest binding against *E. c. carinatus* venom in comparison to the other antivenoms (*p* ≤ 0.0015). As a result, this antivenom was down-selected for in vivo assays. The in vitro selection of antivenom for in vivo assays was undertaken to minimise the use of experimental animals.

#### 2.4.2. In Vivo Neutralisation Potential

While in vitro assays underscore the immunorecognition capability of antivenoms, they cannot definitively predict neutralisation potency. Therefore, we employed the murine model of envenoming to determine the neutralising potency of commercial Indian polyvalent antivenoms against *E. c. carinatus* venom. In vitro experiments identified the Premium Serums antivenom as having relatively better immunological reactivity against *E. c. carinatus* venom and, hence, was down-selected for in vivo testing. We injected groups of mice (*n* = 5) with five graded amounts of venom to determine the LD_50_ of the venom, which was found to be 0.753 mg/kg (Appendix A). The median lethal dose of *E. c. carinatus* from the coastal region of Karnataka was in the range of LD_50_s reported for this sub-species [10,11]. In the neutralisation assays, mice were challenged with a dose of 5× LD_50_ and five different amounts of the antivenom to determine the neutralising potency (Table 1). The neutralising potency (1.39 mg/mL) was approximately four times higher than the manufacturer’s claim (0.45 mg/mL).

Our research demonstrated that the commercial Indian antivenom exhibits appreciable binding efficacy and neutralising potency against the venom of *E. c. carinatus* from Karnataka, which is consistent with findings against other populations [10,11]. Although the reported neutralising potencies were much higher than the marketed claim, this should be further assessed against the pan-Indian populations of *E. c. carinatus* spanning diverse biogeographies. Moreover, as *E. c. carinatus* venom is also known to cause devastating local effects, antivenom efficacy should also be tested against the morbid effects of the venom.

## 3. Conclusions

In conclusion, this study utilised venom proteomics and transcriptomics, along with various biochemical assays and in vitro and in vivo experiments, to uncover the composition and toxicity of *E. c. carinatus* venom. Our findings revealed that the venom of this sub-species is primarily composed of SVMPs, CTLs, LAAOs, SVSPs, and PLA_2_s. The description of the venom gland transcriptome led to the identification of a major protease inhibitor (SVMPI) in the proteome of *E. c. carinatus* for the first time. Moreover, the transcriptome profile highlighted the differential regulation of SVSP and PLA_2_ at the translational level. Given the limited preclinical or clinical evidence highlighting the usefulness of Indian polyvalent antivenom in treating *E. c. carinatus* envenoming, we evaluated its effectiveness under in vitro and in vivo conditions. These preclinical assays demonstrated that the neutralising potency of Indian antivenoms exceeded the marketed claims. However, the biochemical characterisation revealed a significant intra-population variation in proteolytic, PLA_2_, and LAAO activities, which could affect the clinical outcomes and the neutralising capability of commercial antivenoms. Finally, the neutralising potency of the antivenom remains to be evaluated against the morbid effects of *E. c. carinatus* envenoming. Overall, our findings highlight the necessity for investigations into both inter- and intra-population venom variation in *E. c. carinatus*, as well as the impact of this variation on the neutralising efficacy of Indian antivenoms.

## 4. Materials and Methods

### 4.1. Venoms and Antivenoms

Venoms were collected from four (one male: EcCaKA07 and three females: EcCaKA08, EcCaKA09, and EcCaKA10) individuals of *E. c. carinatus* from the Kumta region of Karnataka with prior permission from the State Forest Department (Letter #: PCCF(WL)/E2/CR-06/2018-19, dated 23 February 2022 and Letter #: PCCF/WL/E2/CR-06/2018-19, dated 8 January 2024). The collected venoms were promptly flash-frozen in liquid nitrogen. This was lyophilised and stored at −80 °C until further use. On the fourth day post-venom extraction, two *E. c. carinatus* individuals from two different populations (EcCaKA05 from Dharwad; EcCaKA22 from Kumta) were humanely euthanised, followed by the surgical extraction of both venom glands (left: VGL; right: VGR) and physiological tissues. The antivenoms analysed in this study were manufactured by Bharat Serums and Vaccines Ltd. (Navi Mumbai, India), Haffkine BioPharmaceutical Corporation Ltd. (Mumbai, India), Premium Serums and Vaccines Pvt. Ltd. (Mumbai, India), VINS Bioproduct Ltd. (Thimmapur, India), and Biological E Ltd. (Hyderabad, India). Detailed product information for these antivenoms can be found in Appendix A.

### 4.2. Ethical Statement

The median lethal dose (LD_50_) of venom and median effective dose (*ED*_50_) of antivenom were carried out using CD-1 mice (18–22 g, male) at the Central Animal Facility, Indian Institute of Science, Bangalore (Registration number: 48/GO/ReRcBiBt-S/Rep/99/CPCSEA 18-05-2022). All experiments followed the World Health Organisation (WHO) protocols. Ethical clearance was obtained from the Institutional Animal Ethics Committee (IAEC) of the Indian Institute of Science (IISc), Bangalore (CAF/Ethics/813/2021). Handling of the animals was performed following the guidelines set forth by the Committee for the Purpose of Control and Supervision of Experiments on Animals (CPCSEA). Mice were euthanised by CO_2_ asphyxiation by trained personnel post-experiment or upon reaching humane endpoints, whichever was earlier.

### 4.3. Comparative Tissue Transcriptomics

#### 4.3.1. RNA Isolation, Library Preparation, and Sequencing

The left and right venom glands (VGL and VGR) and physiological tissues (muscle or intestine) were collected from two *E. c. carinatus* individuals (EcCaKA22 and EcCaKA05) in TRIzol™ Reagent (Invitrogen, Thermo Fisher Scientific, Waltham, MA, USA) and stored at −80 °C. The total RNA was isolated using TRIzol™ reagent following the manufacturer’s instructions. The purity of the isolated RNA was measured using an EPOCH 2 microplate spectrophotometer (BioTek Instruments, Inc., Winooski, VT, USA) and a Nanophotometer (Implen NanoPhotometer^®^, Munich, Germany). The integrity of the RNA was then estimated on an Agilent Bioanalyzer 2100 (Agilent Technologies, Santa Clara, CA, USA), and samples with high RNA integrity numbers (RIN > 8) were selected for library preparation and sequencing. Then, cDNA libraries were generated using an NEBNext^®^ Ultra™ RNA Library Prep Kit (New England Biolabs, Ipswich, MA, USA) and sequenced on an Illumina Novaseq 6000 platform with 60 million reads (2 × 150 bp; paired-end) per sample.

#### 4.3.2. Transcriptome Assembly, Annotation, and Quantification

The transcriptome data were curated using Trimmomatic [22] by removing adapter sequences, low-quality leading and trailing (<3) bases, short reads (<20 bases), and low-quality reads (<25, sliding window). The quality of the samples was verified pre- and post-trimming using FastQC [23]. The trimmed data were then de novo assembled using Trinity with the default settings [24]. The completeness of the assembly was assessed using the Sauropsida dataset in BUSCO [25], and the quality of the assembly was evaluated by mapping the reads back onto the assembly using BowTie2 [26]. The coding regions were predicted from the contigs using TransDecoder [27] and were annotated using a BLAST search (v.2.16.0) [28] against the National Center for Biotechnology Information non-redundant (NCBI-NR) database. The abundance of the transcripts was calculated using the RSEM package [29], and the differentially expressed transcripts were identified using a novel non-parametric method with the NOIseq statistical package [30].

### 4.4. Venom Proteomics

#### 4.4.1. Protein Estimation and One-Dimensional Gel Electrophoresis

The total protein concentration of the *E. c. carinatus* venom was quantified using the Bradford assay, with bovine serum albumin (BSA) serving as the standard [31]. Subsequently, the venom was analysed both qualitatively and quantitatively by sodium dodecyl sulphate—polyacrylamide gel electrophoresis (SDS-PAGE) [32]. The proteins present in venoms were electrophoretically separated by using a 15% gel in Tris–Glycine–SDS (TGS) buffer (Cat no. 14374, SRL, Mumbai, India). A total of 12 μg of venom was loaded and run at 80 V. The Precision Plus Protein Dual Colour Standard (Bio-Rad) was used as a marker ladder. The gel was stained with Coomassie Brilliant Blue R-250 (Sisco Research Laboratories Pvt. Ltd., Mumbai, India), followed by visualisation in the iBright CL1000 gel documentation system (Thermo Fisher Scientific, Waltham, MA, USA).

#### 4.4.2. Reversed-Phase High-Performance Liquid Chromatography (RP-HPLC)

The *E. c. carinatus* (from the individual EcCaKA08) venom was fractionated on a Shimadzu CBM-20A RP-HPLC System, following a previously described protocol [33]. Briefly, 1 mg of venom diluted in 100 μL of HPLC grade water was loaded on a Shim-pack GIST C_18_ column (Shimadzu P.No. 227-30017-08; 4.6 mm × 250 mm) with a particle size of 5 μm. A constant flow rate of 1 mL/min was maintained under the following gradient of buffer A [0.1% trifluoroacetic acid (TFA) in HPLC grade water] and buffer B (0.1% TFA in acetonitrile): 5% B for 5 min, followed by 5–15% B for 10 min, 15–45% B for 60 min, and 45–70% B for 10 min during elution of different fractions. Furthermore, 70% of solution B was run for 9 min, followed by 70–98% of solution B for 6 min. Finally, equilibration was performed with 5% of solution B for 10 min. Protein concentration was observed by measuring absorbance at 215 nm using an SPD-M20A Photodiode Array (PDA) detector.

#### 4.4.3. In-Gel and In-Solution Digestion and Liquid Chromatography—Tandem Mass Spectrometry (LC-MS/MS)

The obtained venom fractions were separated using 12.5% SDS-PAGE. Distinct bands were cut and processed by in-gel digestion. It was then reduced at 56 °C for 1 h using 10 mM dithiothreitol (DTT). The excised gel bands were subjected to alkylation in the dark at room temperature (RT) with 55 mM iodoacetamide (IAA) for 45 min. The bands were then washed with 50 mM ammonium bicarbonate and dehydrated using a 100% acetonitrile solution. A vacuum concentrator was used to remove the excess solvent (Thermo Fisher Scientific, Waltham, MA, USA). The samples were digested by 2 ng/μL trypsin (trypsin: protein in a 1:60 ratio; sequencing grade from bovine pancreas, Merck) overnight at 37 °C. The extraction of digested peptides employed 300 μL of 70% acetonitrile solution. In-solution fractions underwent sequential reduction using DTT (100 mM, D5545-Sigma Aldrich, St. Louis, MI, USA) and alkylation via IAA (100 mM, 16125-Sigma), followed by tryptic digestion at a 1:60 trypsin–protein ratio.

These digested peptides were analysed by LC-MS/MS using a quadrupole time of flight (Q-TOF) analyser (Maxis impact, Bruker Daltonics, Bremen, Germany) coupled to an HPLC system (1260 Infinity, Santa Clara, CA 95051, USA). It was equipped with an SB-C18 column (P.N. 683975-902; 4.6 × 150 mm) of 2.7 μm particle size and 120 Å pore size. A 15 μL sample injection was followed by gradient elution of buffer A [0.1% formic acid in MS grade water] and buffer B [0.1% formic acid in acetonitrile (MS grade) solutions] over 60 min at a constant flow rate of 0.2 mL/min. For elution, solution B was used in an increasing gradient from 5% to 95% over 55 min. The mass spectrometry analysis was operated in positive ion mode. The MS scan covered a range of 50–3000 m/z at a spectral rate of 2.00 Hz, with the source temperature held at 220 °C and capillary voltage at 3500 V. Precursor ions exceeding a threshold of 1000 counts were selected for MS/MS fragmentation, and it was performed using a collision cell with collision-induced dissociation (CID) scanning from 400 to 1900 m/z.

Toxin family identifications within fractions and gel bands were performed using PEAKS Studio 11 (Bioinformatics Solutions Inc., Waterloo, ON, Canada). The raw MS/MS data were searched against the NCBI-NR Serpents database (taxid: 8570; 4 April 2025) and the two transcriptomes that were made in this study. Search parameters included the following: parent and fragment mass error tolerance limits of 10 ppm and 0.6 Da, respectively; ‘monoisotopic’ precursor ion search type; ‘specific’ trypsin digestion; carbamidomethylation (+57.02) as a fixed modification; oxidation (+15.99) as a variable modification; false discovery rate (FDR) of 0.1; discovery of ≥1 unique peptide; and a -10log P protein score of ≥50. The raw spectrometry data is accessible via the online depository of ProteomeXchange Consortium (PRIDE partner repository) with the data identifier PXD064002.

#### 4.4.4. Relative Abundance of Toxin Families

The venom proteome quantification used two pre-MS and one post-MS decomplexation step [34]. For the pre-MS decomplexation, the relative abundance of fractions was estimated by normalising the HPLC peak area (*AUC_HPLC_*), followed by densitometric analysis of multiple protein bands (*D_i_*) resolved via SDS-PAGE. This is shown as follows:$$F_{i}={AUC}_{HPLC}\times D_{i}$$
where *F_i_* is the relative abundance of the *i*th fraction

Post-MS decomplexation involved assessing each fraction using tandem mass spectrometry (MS/MS) and normalising the peak areas of MS1 peptide ions for the given toxins (*AUC_MS_*). Detected toxins were categorised into their respective toxin families. The relative abundance of each toxin family was then measured by adding the abundances of all toxins allocated to that family across all fractions. The calculation was executed using the following equation, where *X_j_* represents the *j*th toxin family, ‘*N*’ denotes the total number of toxins in the family, and *i* corresponds to the fraction number:$$elative\;abundance\;of\;X_{j}\;(\%)=\sum_{j=1}^{N}F_{i}\;\times \;{AUC}_{MS}$$

### 4.5. Biochemical Characterisation

#### 4.5.1. Phospholipase Assay

The phospholipase activity of *E. c. carinatus* venoms was assessed following a previously described method [33] using a chromogenic lipid substrate, 4-nitro-3-[octanoyloxy] benzoic acid (NOB; Enzo Life Sciences, New York, NY, USA). Briefly, 5 μg of individual venoms were incubated at 37 °C with 100 μL of 500 μM NOB substrate dissolved in a 200 μL reaction buffer (10 mM Tris–HCl, 10 mM CaCl2, 100 mM NaCl, pH 7.8). The assay kinetics were observed over 40 min by measuring absorbance at 425 nm every 10 min with an Epoch 2 microplate spectrophotometer (BioTek Instruments, Inc., USA). A standard curve was plotted with varying concentrations of the NOB substrate (4 nmol to 130 nmol) and 4 M NaOH using an identical protocol, from which the amount of the phospholipid substrate in nmol cleaved per minute per mg of the venom was calculated.

#### 4.5.2. Protease Assay

The proteolytic activities of *E. c. carinatus* venoms were assessed using a method that had been previously described [33]. A total of 10 μg of each venom was incubated with 80 μL of 5 mg/mL of the substrate (azocasein) at 37 °C for 90 min, followed by the addition of trichloroacetic acid (200 μL) to end the reaction. After centrifuging this mixture at 1000× *g* for 5 min, an equal volume of 0.5 M NaOH was added to the supernatant. The absorbance at 440 nm was then measured in an Epoch 2 microplate spectrophotometer (BioTek Instruments, Inc., USA). Purified bovine pancreatic protease (Sigma-Aldrich, USA) served as the positive control, and the relative proteolytic activity was calculated. A fixed amount (5 μL) of 100 mM ethylenediamine tetraacetic acid (EDTA) and 40 mM phenylmethylsulfonyl fluoride (PMSF) were incubated with 10 μg of the venoms to understand the relative contribution of snake venom metalloproteinases (SVMPs) and snake venom serine proteases (SVSPs), respectively, to the proteolytic activities.

#### 4.5.3. L-Amino Acid Oxidase (LAAO) Assay

A previously described protocol was used to evaluate the LAAO activity of *E. c. carinatus* venoms [10], wherein 0.5 μg of each venom was incubated at 37 °C for 10 min. Subsequently, 200 μL of the reaction mixture (5 mM L-leucine, 50 mM Tris-HCl buffer, 5 IU/mL horseradish peroxidase, and 2 mM o-phenylenediamine dihydrochloride) was added, followed by a further 10 min incubation at 37 °C. Post-incubation, the reaction was stopped by adding 50 μL of 2 M H_2_SO_4_ solution. The absorbance was recorded at 492 nm using an Epoch 2 microplate spectrophotometer (BioTek Instruments, Inc., USA). A standard curve was generated using an identical protocol by incubating H_2_O_2_ with the reaction mixture, from which the amount of L-leucine (nmol) cleaved by 1 mg of venom per minute was calculated.

### 4.6. In Vitro Binding Assay

To assess the in vitro binding potential of commercial Indian antivenoms towards the venoms from *E. c. carinatus*, a previously established indirect ELISA protocol was followed [33]. Briefly, 100 ng of these venoms were diluted in a carbonate buffer (pH 9.6) and coated onto 96-well plates, followed by incubation at 4 °C overnight. The following day, the unbound venom was removed by washing with Tris-buffered saline in 1% Tween 20 (TBST). The plates were then treated with a blocking buffer (5% skimmed milk in TBST) and incubated at room temperature for 3 h before undergoing a final TBST wash. Post-wash, various dilutions of Indian antivenom (Premium Serums, VINS Bioproduct, Bharat Serums, Haffkine BioPharmaceutical, or Biological E) were added, and plates were incubated overnight at 4 °C. The next day, TBST wash was carried out to remove the loosely or unbound antivenoms, followed by the addition and incubation with secondary horseradish peroxidase (HRP)-conjugated rabbit anti-horse antibodies (Sigma-Aldrich, USA; dilution 1:1000) for 2 h at room temperature. Following incubation, ABTS [2,2′-azino-bis(3-ethylbenzothiazoline-6-sulfonic acid)] substrate solution (Sigma-Aldrich, USA) was added. Absorbance at 405 nm was then measured for 40 min in an Epoch 2 microplate reader. Naive horse IgGs (Bio-Rad Laboratories, USA) served as the negative control, against which the bindings of the antivenoms were compared.

### 4.7. The Median Lethal Dose (LD_50_)

The toxicity of *E. c. carinatus* venom was estimated using a murine model of snake envenoming. Here, five venom concentrations made in 0.9% saline were injected intravenously (tail vein) into male CD-1 mice (18–22 g; n = 5/dose group). Injected mice were monitored over a period of 24 h, and the symptoms were documented. Following this observation time, the quantities of deceased and surviving mice in each dosage group were recorded, and Probit analysis was employed to ascertain the LD_50_ [35].

### 4.8. The Median Effective Dose (ED_50_)

Among all the tested antivenoms, the Premium Serums’ antivenom was down-selected (possessing the highest venom recognition potential indicated in indirect ELISA experiments) for the venom neutralisation experiments that followed the WHO-recommended protocol. Briefly, four dilutions of antivenom were mixed with a challenge dose (5× LD_50_) of venom and incubated at 37 °C for 30 min, followed by the intravenous injection into the tail vein of male CD-1 mice (18–22 g; 5 mice per group). The signs and symptoms of envenoming were recorded, and the numbers of dead and surviving mice were noted after a 24 h observation period. The *ED*_50_ was estimated using Probit statistics, while the neutralisation potency was calculated using the formula below, where ‘*n*’ denotes the number of *LD*_50_ used as the challenge dose:(1)$$Antivenom\;neutralisation\;potency\;(mg/mL)=\frac{\left(n-1)\;\times \;{LD}_{50}\;of\;venom\;(mg/mouse\right)}{{ED}_{50}\;\;(mL)\;}$$

### 4.9. Statistics

For statistical analyses, one-way and two-way ANOVA were implemented along with Tukey’s correction for multiple comparisons in GraphPad Prism 8 (GraphPad Software, La Jolla, CA, USA, www.graphpad.com).

## Figures and Tables

**Figure 1 toxins-17-00353-f001:**
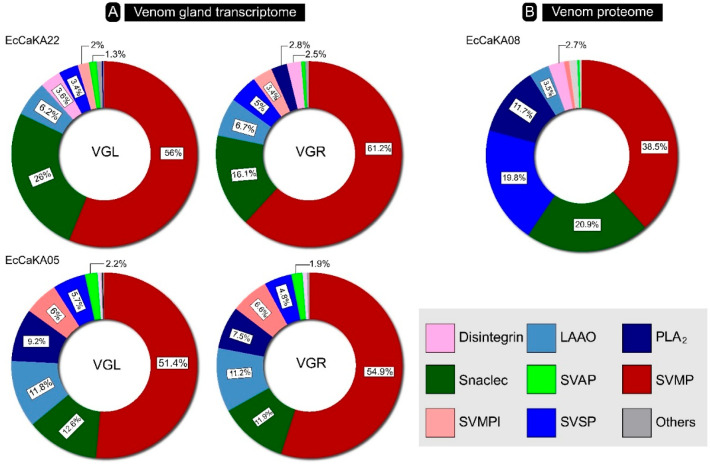
(**A**) The comparative venom gland transcriptomes and (**B**) venom proteome of *E. c. carinatus* from Karnataka are shown as doughnut charts. Each toxin family is depicted in a unique colour, and the relative abundances are mentioned in percentages. LAAO: L-amino acid oxidase, PLA_2_: phospholipase A_2_, Snaclec: snake venom C-type lectin, SVAP: snake venom aspartic protease, SVMP: snake venom metalloprotease, SVMPI: snake venom metalloprotease Inhibitor, SVSP: snake venom serine protease.

**Figure 2 toxins-17-00353-f002:**
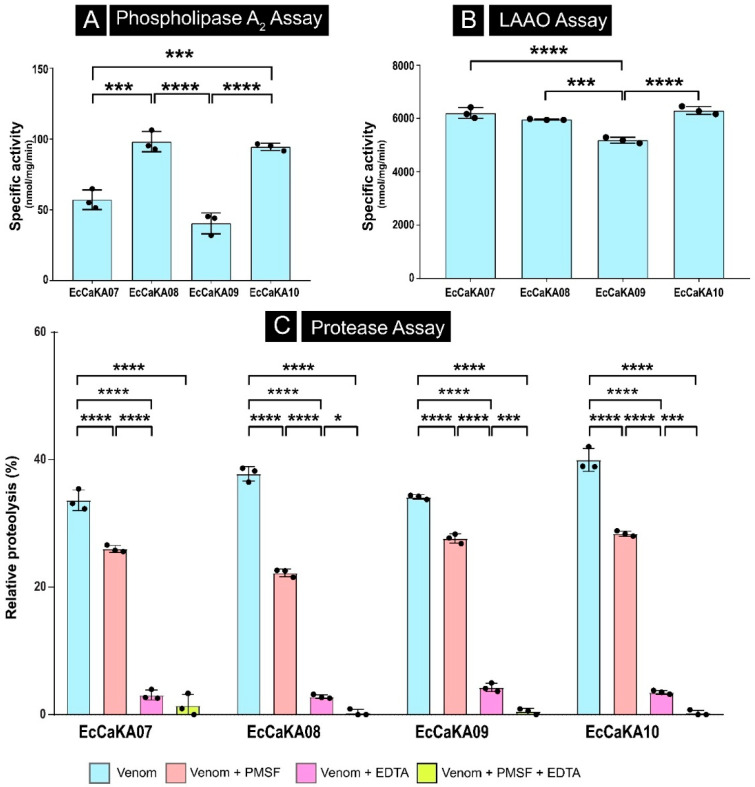
The biochemical activities of *E. c. carinatus* venom. (**A**) phospholipase A_2_ activity; (**B**) LAAO activity; and (**C**) protease activity (with and without SVSP and/or SVMP inhibition) are depicted here. All assays were performed in triplicate, and the standard deviation is represented as error bars. Here, * *p* < 0.05; *** *p* < 0.001; **** *p* < 0.0001 (one-way ANOVA for Figure 2A,B; two-way ANOVA for Figure 2C).

**Figure 3 toxins-17-00353-f003:**
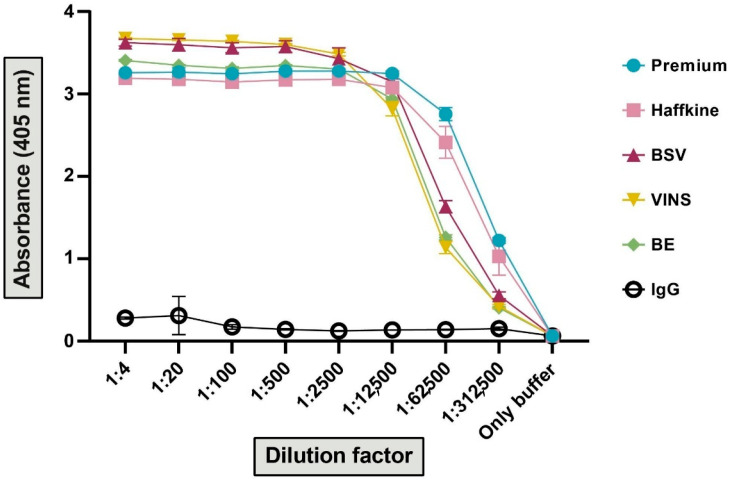
The binding efficacy of the Indian polyvalent antivenoms against *E. c. carinatus* venom. The absorbance at 405 nm (y-axis) is plotted against various dilutions of antivenoms (x-axis). Each antivenom is uniquely represented by a distinct colour and shape. The values are provided as the mean absorbance of triplicates, and the error bars represent the standard deviation.

**Table 1 toxins-17-00353-t001:** The median effective dose and neutralisation potency of Premium Serums’ antivenom against *E. c. carinatus* venom.

Name of Sample	Challenge Dose 5× *LD*_50_ (μg/mouse)	Amount of Antivenom Injected in the Venom-Antivenom Mixture (µL)				*ED*_50_ (μL)	Potency of Antivenom (mg/mL)
EcCaKA (pooled)	75	39.1	45	51.75	68.4	43.07 39.7–46.72	1.39 1.28–1.51

## Data Availability

The raw proteome data can be found in the PRIDE repository with the data identifier PXD064002. The raw transcriptomics data described in this study can be openly accessed via the Sequence Read Archive (SRA) at NCBI (Bioproject: PRJNA1273671; SRA: SRR33884852, SRR33884853, SRR33884854, SRR33884855, SRR33884856, and SRR33884857).

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
