# Peer review of "Explaining Echis: Proteotranscriptomic Profiling of Echis carinatus carinatus Venom"

_toxins, 2025, doi:10.3390/toxins17070353_

Round 1
Reviewer 1 Report
Comments and Suggestions for Authors
Review of the article: Explaining Echis: Proteotranscriptomic profiling of Echis carinatus venom
Manuscript ID: toxins-3713250
The author(s) employed venom proteomics and transcriptomics, along with various biochemical assays and both in vitro and in vivo experiments, to elucidate the composition and toxicity of E. carinatus venom. The study offers a comprehensive analysis of its venomics and antivenomics, addressing a significant knowledge gap concerning the 'big four' Indian snakes and contributing valuable insights for snakebite treatment in India. I have only a few comments regarding the methods, statistical analysis, figure presentation, text editing, and supplementary materials. Once these are addressed, I believe the manuscript will be suitable for publication in Toxins.
- Lines 60–61: Please clarify the precise collection locations of the six snakes. Why are they considered to belong to the same “group”? It may be helpful to provide a map showing the collection sites.
- Line 66: The conventional protocol is to milk venom a few days before surgically extracting the venom glands in order to maximise mRNA expression. Could you explain why this standard approach was not followed?
- Line 263: It is essential to verify assumptions of normality and homoscedasticity before conducting parametric analyses such as ANOVA. I recommend including both the test statistics (e.g., F-values with degrees of freedom) and p-values throughout the manuscript, including results for normality and homoscedasticity tests.
- Line 287: I suggest replacing the general term “C-type lectin” with “snaclecs,” which specifically refers to C-type lectins found in snake venom.
- Lines 304–305: Please discuss why the translation levels of SVSPs and PLAâ‚‚s were approximately four times and 1.5–5 times higher, respectively.
- Line 318: Please spell out the full name of “SVAP” for clarity.
- Figure 2A and 2B: The Y-axis label “Speicific” should be corrected to “Specific.”
- Lines 387–388: Please clarify how in vitro selection of antivenom can reduce or minimise the use of animals in in vivo assays.
- Line 404: Should “five” be revised to “four”? Please double-check for consistency.
- Line 406: The manufacturer’s neutralising potency claim (0.45 mg/mL) refers to which specific antivenom product (Premium Serums, VINS Bioproduct, Bharat Serums, Haffkine BioPharmaceutical, or Biological E)?
- Supplementary Figure 1A: Which specific peaks correspond to the labels (e.g., Peak 1, 2, 4, 15, or 16)? Also, were any peaks excluded from mass spectrometry analysis, and if so, could this introduce bias into the proteomic results?
- Supplementary Figure 1B: It is stated that each peak was divided into up to four bands, but some peaks show more than four visible bands. Please revise the description for accuracy.
- Supplementary Tables 3 and 5: Please explain what the ranges such as “10.3–22.02,” “0.515–1.101,” “39.7–46.72,” and “1.28–1.51” represent.
- Supplementary Table 4: It appears that some antivenom samples used in the study were past their expiration date. Please clarify why expired materials were used and whether this could have influenced the outcomes.
Author Response
Thank you very much for taking the time to review this manuscript. We found your comments valuable. Please find the detailed responses below and the corresponding revisions highlighted in the resubmitted files.
Comment 1: Lines 60–61: Please clarify the precise collection locations of the six snakes. Why are they considered to belong to the same “group”? It may be helpful to provide a map showing the collection sites.
Response 1: We collected venom samples from six individuals of Echis carinatus in the region of Kumta, Karnataka. In this study, we used venom samples from four individuals, as the venom yield obtained from the remaining two was very low. Since all the venom samples were collected from a single location in Kumta, Karnataka, India, a map was not provided. We have used one sample from Dharwad, Karnataka, for transcriptome (EcCaKA05). But no venom from this individual was used for analyses, as it produced very little of it. We have now clarified this in the manuscript (Line 238):
“Venoms were collected from four (one male and three females) individuals of E. carinatus from the Kumta region of Karnataka…”.
Comment 2: Line 66: The conventional protocol is to milk venom a few days before surgically extracting the venom glands in order to maximise mRNA expression. Could you explain why this standard approach was not followed?
Response 2: We have followed the standard approach of milking the venom 3 days before the extraction of the venom gland. The surgical extraction of the venom gland took place on the 4th day. We have now clarified this in the text (Line no. 243-246):
“On the fourth day post-venom extraction, two E. carinatus individuals (EcCaKA05 from Dharwad; EcCaKA22 from Kumta) were humanely euthanised, followed by the surgi-cal extraction of both venom glands (left: VGL; right: VGR) and physiological tissues.”.
Comment 3: Line 263: It is essential to verify assumptions of normality and homoscedasticity before conducting parametric analyses such as ANOVA. I recommend including both the test statistics (e.g., F-values with degrees of freedom) and p-values throughout the manuscript, including results for normality and homoscedasticity tests.
Response 3: F-values with degrees of freedom are added to the manuscript. Quantile-Quantile plots and Homoscedasticity plots were generated for each dataset, and the normality was verified. We haven’t included the graphs as we believe they don’t add much to the results, but they are now made accessible to the reviewer for review.
Comment 4: Line 287: I suggest replacing the general term “C-type lectin” with “snaclecs,” which specifically refers to C-type lectins found in snake venom.
Response 4: We have changed it in the manuscript.
Comment 5: Lines 304–305: Please discuss why the translation levels of SVSPs and PLAâ‚‚s were approximately four times and 1.5–5 times higher, respectively.
Response 5: Thanks for the comment. The relative abundance of PLA2 showed a large variation between the two individual transcriptomes we have profiled. Thus, we have provided a range. Since we haven’t looked at the regulatory aspects of the translation and transcription, we believe it is out of scope for the paper to discuss the reasons. However, following the suggestion from the reviewer, we have now added a line on a possible differential regulation on the translational level: (Line 98-99):
“This could point to a differential regulation for these toxins at the level of translation”.
Comment 6: Line 318: Please spell out the full name of “SVAP” for clarity.
Response 6: Added this to the manuscript.
Comment 7: Figure 2A and 2B: The Y-axis label “Speicific” should be corrected to “Specific.”
Response 7: Thank you for pointing this out. We have changed it in the manuscript.
Comment 8: Lines 387–388: Please clarify how in vitro selection of antivenom can reduce or minimise the use of animals in in vivo assays.
Response 8: By utilising the in vitro selection method, we selected the best binding antivenom for in vivo assays, significantly reducing animal usage. Instead of determining the antivenom potency for five different antivenoms, which would require 125 mice (25 mice per antivenom), we focused on calculating the potency of a single antivenom using just 25 mice.
Comment 9: Line 404: Should “five” be revised to “four”? Please double-check for consistency.
Response 9: Yes, thank you for pointing this out. We have changed it in the manuscript. (Line no. 432)
Comment 10: Line 406: The manufacturer’s neutralising potency claim (0.45 mg/mL) refers to which specific antivenom product (Premium Serums, VINS Bioproduct, Bharat Serums, Haffkine BioPharmaceutical, or Biological E)?
Response 10: The neutralising potency of all Indian antivenoms is 0.45 mg/mL, but in this case, it specifically refers to Premium Serums.
Comment 11: Supplementary Figure 1A: Which specific peaks correspond to the labels (e.g., Peak 1, 2, 4, 15, or 16)? Also, were any peaks excluded from mass spectrometry analysis, and if so, could this introduce bias into the proteomic results?
Response 11: The image was edited to remove the ambiguity for the peaks mentioned. No peaks before 100 min were excluded.
Comment 12: Supplementary Figure 1B: It is stated that each peak was divided into up to four bands, but some peaks show more than four visible bands. Please revise the description for accuracy.
Response 12: Thank you for pointing this out. We have modified accordingly: “The SDS-PAGE for the eight major peaks of E. carinatus venom after RP-HPLC is depicted here. Each lane was excised as shown above (marked as a, b, c, or d) and subjected to in-gel digestion before mass spectrometry.”
Comment 13: Supplementary Tables 3 and 5: Please explain what the ranges such as “10.3–22.02,” “0.515–1.101,” “39.7–46.72,” and “1.28–1.51” represent.
Response 13: These are the 95% fiducial confidence intervals of the respective LD50 or ED50 values.
Comment 14: Supplementary Table 4: It appears that some antivenom samples used in the study were past their expiration date. Please clarify why expired materials were used and whether this could have influenced the outcomes.
Response 14: In our experience, and as demonstrated by other groups, antivenoms do not truly expire when stored properly. We aimed to avoid wasting a life-saving product for preclinical testing, given that antivenoms often face shortages in many regions across India. Furthermore, it is evident that the "expiry" date does not impact the neutralisation potency. The tested antivenom exhibited significantly greater neutralisation potency despite being past its expiry date, even surpassing the marketed value.
Reviewer 2 Report
Comments and Suggestions for Authors
The reviewed article presents an attempt at a comprehensive analysis of the composition of Echis carinatus venom and the transcripts from which these components are derived. The article is interesting and potentially important for the development of venomics, but it has several shortcomings that I believe should be corrected. I have included a list of my comments below.
- Please clarify whether the individuals from which material was collected for transcriptomics studies were the same ones from which venom was collected? What was the interval between venom collection and gland isolation for transcriptomics?Please add in this section information about the age of the snakes and whether they were caught from the wild and the venom was collected there, or whether they were caught and later lived in captivity.
- I have always been taught that CBB R-250 cannot be used for densitometry because it has a very small dynamic range. Please comment on this issue.
- The methodology says that venoms were analyzed separately (e.g., in line 191), so why are results from 4 individuals presented and not 6 since venom was collected from so many?
- Table S2 needs to be expanded, please complete it with protein names, peptide sequences and information on how many and which peptides were unique. This is very important especially in the context of SVMPI and SVAP proteins.
- I cannot agree with the sentence in line 300 “The venom proteome of E. carinatus was largely consistent with the literature [7,8,10], as well as our venom gland transcriptome.” Anyway, the authors themselves don't seem to agree with this because below they themselves describe the differences, which I feel are clear.
- Question for SF1B: why were peak 1 and especially peak 3 not separated on the gel?
- In line361, the authors emphasize how important it is that they performed biochemical analyses that detected venereal differences. This is inconsistent given that the main proteomics experiment was performed on the pooled venom.
- There is no explanation of the abbreviation SVAP in the caption of Figure 1
- I suggest that Table S5 be placed in the main text of the article.
- Please add in the discussion information whether the antivenoms used were polyvalent and whether, according to the manufacturer, they should work on E. carinatus
- The sentence “Additionally, the description of the venom gland transcriptome led to the identification of a major protease inhibitor (SVMPI) in the proteome of E. carinatus...” is a mental abbreviation. The presence of a protein cannot be determined from the transcriptome.
- Please ensure that the entire text of the article is written in the same font (e.g., line 62, 74, 386) and all Latin names including “in vitro” and “in vivo” are written in italics.
Author Response
Thank you very much for taking the time to review this manuscript. We found your comments valuable. Please find the detailed responses below and the corresponding revisions highlighted in the resubmitted files.
Comment 1: Please clarify whether the individuals from which material was collected for transcriptomics studies were the same ones from which venom was collected? What was the interval between venom collection and gland isolation for transcriptomics? Please add in this section information about the age of the snakes and whether they were caught from the wild and the venom was collected there, or whether they were caught and later lived in captivity.
Response 1: Thanks for the comment. The Individuals used for transcriptomics and proteomics are different. The corresponding IDs of the individuals are provided in the manuscript. We collected venoms from four adult snakes that were caught in the wild on the Kumta Plateau in Karnataka. After extracting the venom, we released three of the snakes back into the wild. After venom extraction, one snake from the Kumta region and another from the Dharwad region (both in Karnataka) were housed in a secure enclosure at the respective Forest Department for three days. On the fourth day, with assistance from an expert veterinarian, we humanely euthanised this snake and performed a venom gland extraction. These are very small snakes, and most often it's very difficult to collect ‘enough’ venom for rigorous proteomics, biochemical, pharmacological and toxinological analyses. The snakes that we euthanised didn’t provide enough venom for proteomics analysis. (Line no. 243-246):
“On the fourth day post-venom extraction, two E. carinatus individuals (EcCaKA05 from Dharwad; EcCaKA22 from Kumta) were humanely euthanised, followed by the surgi-cal extraction of both venom glands (left: VGL; right: VGR) and physiological tissues.”.
Comment 2: I have always been taught that CBB R-250 cannot be used for densitometry because it has a very small dynamic range. Please comment on this issue.
Response 2: Thanks for the comment. But since we are using densitometry results to calculate relative abundance measurements, and not for absolute values, the use of CBB R-250 is justified.
Comment 3: The methodology says that venoms were analyzed separately (e.g., in line 191), so why are results from 4 individuals presented and not 6 since venom was collected from so many?
Response 3: Thanks for pointing this out. Although the venom was collected from 6 individuals, we could only use 4 venoms in this study, as the venom yield obtained from two of the snakes was very low. We have edited the text accordingly. (Line 238):
“Venoms were collected from four (one male and three females) individuals of E. carinatus from the Kumta region of Karnataka…”.
Comment 4: Table S2 needs to be expanded, please complete it with protein names, peptide sequences and information on how many and which peptides were unique. This is very important especially in the context of SVMPI and SVAP proteins.
Response 4: Thanks for the comment. We take that the reviewer is referring to File S2, which has proteome data, and have edited accordingly.
Comment 5: I cannot agree with the sentence in line 300 “The venom proteome of E. carinatus was largely consistent with the literature [7,8,10], as well as our venom gland transcriptome.” Anyway, the authors themselves don't seem to agree with this because below they themselves describe the differences, which I feel are clear.
Response 5: Thanks for the comment. In the initial sentence, we were trying to point out the overall consistency in terms of major toxins present. Edited to fit this clarification. (Line no. 92-94)
“The venom proteome of E. carinatus was largely consistent with the literature [7,8,10], as well as our venom gland transcriptome in terms of major toxins present”.
Comment 6:Question for SF1B: why were peak 1 and especially peak 3 not separated on the gel?
Response 6: Peaks 1 and 3 are not considered for SDS PAGE, as their abundance was very small (less than 15 micrograms), and the corresponding band in SDS-PAGE may not be visible with Coomassie brilliant blue stain. So it was taken for Mass spectrometry directly by the in-solution digestion method.
Comment 7: In line361, the authors emphasize how important it is that they performed biochemical analyses that detected venereal differences. This is inconsistent given that the main proteomics experiment was performed on the pooled venom.
Response 7: Please note that the proteomics experiment was not performed on pooled venom, but on an individual venom (EcCaKA08). This is now added in the manuscript (line no. 302) for clarification. The pooled venom was used for determining the LD50 and ED50 values because of limitations with the amount of venom these snakes produce.
Comment 8: There is no explanation of the abbreviation SVAP in the caption of Figure 1
Response 8: Edited in the manuscript.
Comment 9: I suggest that Table S5 be placed in the main text of the article.
Response 9: As per this suggestion, we have added this table to the main text. (Line no. 204)
Comment 10: Please add in the discussion information whether the antivenoms used were polyvalent and whether, according to the manufacturer, they should work on E. carinatus
Response 10: Thank you for pointing this out. We have added this information in the manuscript. (Line no. 175, 186, 194, 225)
Comment 11: The sentence “Additionally, the description of the venom gland transcriptome led to the identification of a major protease inhibitor (SVMPI) in the proteome of E. carinatus...” is a mental abbreviation. The presence of a protein cannot be determined from the transcriptome.
Response 11: Thanks for the comment. What we have meant by this is “the addition of a newly described transcriptome to the database while annotating the proteome has helped us identify the protein (SVMPI) present in the proteome correctly”. We have found SVMPI in both the transcriptome and the proteome. We understand the confusion and have thus edited the text accordingly. (Line no. 220-223 )
“The description of the venom gland transcriptome led to the identification of a major protease inhibitor (SVMPI) in the proteome of E. carinatus for the first time. Moreover, the transcriptome profile highlighted the differential regulation of SVSP and PLA2 at the translational level.”
Comment 12: Please ensure that the entire text of the article is written in the same font (e.g., line 62, 74, 386) and all Latin names including “in vitro” and “in vivo” are written in italics.
Response 12: Thank you for pointing this out. We had noticed that in the journal Toxins, in vitro and in vivo were not written in italics. Regardless, we have changed it in the manuscript as per your suggestion. The font issue is rectified.
Round 2
Reviewer 2 Report
Comments and Suggestions for Authors
I am fully satisfied with the answers to the questions and changes made by the authors in the article.
Author Response
Comment 1: I am fully satisfied with the answers to the questions and changes made by the authors in the article.
Response 1: We thank you for your time and valuable comments.